# Microbes vs. Nematodes: Insights into Biocontrol through Antagonistic Organisms to Control Root-Knot Nematodes

**DOI:** 10.3390/plants12030451

**Published:** 2023-01-18

**Authors:** Adil Ameen Bhat, Adnan Shakeel, Sonia Waqar, Zafar Ahmad Handoo, Abrar Ahmed Khan

**Affiliations:** 1Section of Environmental Botany and Plant Pathology, Faculty of Life Sciences, Aligarh Muslim University, Aligarh 202002, India; 2Mycology and Nematology Genetic Diversity and Biology Laboratory, USDA, ARS, Northeast Area, 10300 Baltimore Avenue, Beltsville, MD 20705, USA

**Keywords:** biocontrol, root-knot nematode, antagonist, disease, interactions

## Abstract

Root-knot nematodes (*Meloidogyne* spp.) are sedentary endoparasites that cause severe economic losses to agricultural crops globally. Due to the regulations of the European Union on the application of nematicides, it is crucial now to discover eco-friendly control strategies for nematode management. Biocontrol is one such safe and reliable method for managing these polyphagous nematodes. Biocontrol agents not only control these parasitic nematodes but also improve plant growth and induce systemic resistance in plants against a variety of biotic stresses. A wide range of organisms such as bacteria, fungi, viruses, and protozoans live in their natural mode as nematode antagonists. Various review articles have discussed the role of biocontrol in nematode management in general, but a specific review on biocontrol of root-knot nematodes is not available in detail. This review, therefore, focuses on the biocontrol of root-knot nematodes by discussing their important known antagonists, modes of action, and interactions.

## 1. Introduction

Plant-parasitic nematodes (PPNs) significantly reduce crop yields in both quantity and quality, which is a growing economic concern for the global agricultural industry [1]. The three most economically important groups of PPNs include root-knot nematodes, cyst nematodes, and lesion nematodes, which infect and proliferate on a wide range of plant species [2]. Among these PPNs, root-knot nematodes (*Meloidogyne* spp.) are the main damaging plant-parasitic nematodes. *Meloidogyne* spp. are sedentary endoparasites with a broad host range and exhibit a negative or detrimental effect on many crops, resulting in a significant reduction in the amount and quality of food production [3]. Not only do they reduce yield, but the infected crops also become more prone to bacterial and fungal diseases [4]. They also interact with viral diseases in several ways [5]. In *Solanum khasium*, the root-knot index was higher on plants inoculated with tobacco mosaic tobamovirus than on healthy plants [6]. Inhibitory effects on root-knot nematode *Meloidogyne javanica* were observed in zucchini (*Cucurbita pepo*) infected with watermelon mosaic polyvirus. Virus infection retarded the establishment of these nematodes in the roots as compared with healthy plants [7]. All of these effects, favorable or detrimental, occurred or were more pronounced when nematode inoculation was preceded for 2 to 3 weeks by virus infection. This points to biochemical processes in plant tissues as a decisive mechanism involved in these interactions. This is underlined by results obtained by Alam et al. [8]. They observed antagonistic effects between tomato mosaic tobamovirus and *M. incognita*. When virus infection preceded nematode inoculations, nematodes were suppressed, and when nematodes were the first agent, the virus was inhibited. The changes caused by one pathogen were detrimental to the other. The influence of the host on this type of interaction was demonstrated by Moura and Powell [9]. In two out of three tomato varieties, the egg production of *M. incognita* was significantly increased by the presence of tobacco mosaic tobamovirus, but in the third, it was not. The concomitant presence of nematode and virus in most cases results in a synergistic effect that aggravates the plant damage considerably. Little is known about the physiological or biochemical basis of enhancing or inhibiting nematode development by a virus infection of the host. It has been suggested that changes in protein metabolism caused by nematodes may interact with corresponding changes caused by the virus, resulting in beneficial effects in one case and detrimental effects in another [10]. This view was recently supported by Shower et al. [11]. They showed that the changes in free amino acid levels in sugarcane (*Saccharum* L.) caused by sugarcane mosaic virus were responsible for population changes in various nematodes on sugarcane. The life cycle of root-knot nematode is very complicated, and based on environmental factors such as temperature, moisture and availability of a suitable host, it can take 3 weeks to a few months to complete its life cycle. In order to thrive inside plant roots, *Meloidogyne* spp. may overpower host defense mechanisms. There are various species of root-knot nematodes that parasitize and infest crops in agroecosystems. However, four species, *M. incognita* (Kofoid and White, 1919) Chitwood, 1949, *M. javanica* (Treub, 1885) Chitwood, 1949, *M. hapla* Chitwood, 1949, and *M. arenaria* (Neal 1889) Chitwood, 1949, are economically most important [12]. In agriculture, nematodes result in a loss of about USD 100 billion annually [13]. It has been estimated that root-knot nematode infestations result in a loss of about 15% in annual crops all over the world [14].

Considering the economic losses caused by root-knot nematodes, there is an immediate need to control these plant parasites by suitable methods. So far, various approaches have been used for their control, including chemicals, host plant resistance, crop rotation, soil solarization, antagonistic organisms, etc. Crop rotation is very commonly used for nematode control, but it requires adequate land to produce alternate crops that are non-hosts to the nematodes. Additionally, these alternate crops must bring good profits to the grower. Chemical treatmentshave been found to be very effective in nematode management, but due to their nonavailability, high cost, and short-term effects, resistance developed in nematodes [15]; further, such treatments can kill useful microorganisms, and persistent chemicals may cause serious threats to the ecological balance [16]. Nematicides may be toxic reproductively and carcinogenic in animals [17]. So, due to limitations on the use of nematicides [18,19], it has become essential to develop cost-effective and ecologically friendly agents. In this regard, biocontrol can be a promising alternate to nematicides and traditional methods with an eco-friendly mechanism to combat and suppress the pathogenic root-knot nematodes in crop ecosystems.

## 2. Biocontrol Agents against Root-Knot Nematodes

Microorganisms play a vital role in soil health, plant growth and development, and are crucial for plant disease management [20]. These microorganisms can be exploited as biocontrol agents by using their natural mode of action against several soil-borne pathogens. Biocontrol means the use of advantageous organisms and their products to increase beneficial responses and reduce negative ones and contribute to gross increases in productivity. Bio-agents lead to disease prevention with no or little environmental threat [21]. These agents may last in the soil for long durations. Through biological control, we can check diseasesthrough various mechanisms, such as:Competition: Competition (intraspecific and interspecific), mainly for space, nutrients, water, etc. reduces the growth, activity, and reproduction of the organisms involved [22] or affects nematode fitness.Antibiosis: This can happen when plant compounds are released from the roots into the soil. Bacteria produce and release certain antibiotics or toxins, which may have an undesirable effect on the infective stage of nematodes [23]. Allelochemicals are known to harm plant-parasitic nematodes as well [24].Parasitism: Nematodes are prey for most nematophagous bacteria, which use nematodes as a potential source of nutrients. They can also pierce the cuticle (due to enzymatic action), and kill the nematode host [25].Plant growth promotion: Bio-agents aid in the control of plant diseases by increasing plant development through improved nutrient solubilization, higher nutrient uptake, and nutrient sequestration. Plants with higher nutritional status can tolerate more plant-parasitic nematodes in their roots [26].Induced systemic resistance: Numerous bacterial products create systemic signaling in plants, which can protect the entire plant from diseases induced by various pathogens or help plants become resistant to various pathogenic organisms [27].

A schematic representation of different mechanisms of biocontrol is shown in Figure 1. Even though a large number of biocontrol agents have been revealed as potential pesticides, only a few have been commercialized [28]. Under laboratory conditions, biocontrol agents appear to perform well compared to field conditions, but in field conditions their effect may drop due to interactions with other biotic and abiotic components of the surrounding atmosphere. Through the application of various biocontrol agents, *Meloidogyne* species have been conveniently controlled [29,30]. A broad range of organisms such as bacteria, viruses, fungi, protozoans, insects, mites, and predatory nematodes have been reported as nematode antagonists [31]. Among these, bacterial and fungal antagonists are the important ones and have been investigated over a period of time for their use as biocontrol agents against plant-parasitic nematodes [32]. Apart from natural antagonists, plant extracts can also be used for nematode control, as discussed in Section 2.3.

### 2.1. Bacteria as Biocontrol Agents against Root-Knot Nematodes

Microbial biocontrol agents exploit antagonism through hyper-parasitism and antibiosis to interfere with and inhibit the growth of another pathogen [33]. Glick [34] found that soil bacteria that promote plant growth are found in alliance with the roots/leaves/flowers or any plant tissues and are generally called plant growth-promoting bacteria (PGPB). Both the plant and microbial community can be influenced by bacteria and their products [35]. Various bacterial species have been evaluated for their nematicidal activities. Based on their mechanisms of operation, bacterial antagonists of nematodes can be grouped into rhizobacteria, obligate parasitic bacteria, endophytic bacteria, opportunistic parasitic bacteria, symbiotic bacteria, and parasporal crystal (cry) protein-forming bacteria [25]. Many bacteria such as *Pseudomonas* spp., *Bacillus* spp., *Burkholderia* spp., and others minimize damage in plants, as they build metabolites that then alter nematode behavior, feeding and reproduction [36]. *Bacillus* and *Pseudomonas* occur widely in the natural environment and have shown the highest efficacy for biological control [37]. In the rhizosphere, *Bacillus* spp. and *Pseudomonas* spp. are the prominent opponents of plant pathogens [38]. *Pseudomonas* spp. are largely present in soil and plant root systems. They can take advantage of plant exudates for nutritional purposes, produce certain metabolites and antimicrobial compounds, and can be easily produced in the laboratory [39]. Some of the important bacterial species are summarized in Table 1.

Sharma et al. [40] studied the control of root-knot disease with pseudomonad rhizobacteria filtrate under *in-vitro* and greenhouse conditions. *Pseudomonas jessenii* strain R62 and *Pseudomonas synxantha* strain R81 were used to control root-knot nematode (*Meloidogyne incognita*) on tomato plants. In laboratory conditions, it was found that out of all of the treatments (25%, 50%, 75%, 100%) with R62 and R81, 75%, 100% and all dilutions of R62 + R81 caused 100% mortality of second-stage juveniles (J2). At 25%, no effect was found on J2, and at 50% some mortality was observed. Under greenhouse conditions, by using R62 and R81 collectively, significant variations in plant growth parameters were observed. When the same treatment (R62 + R81) was given to plants under nematode stress, a great increase in plant growth parameters was found in contrast to only nematode-inoculated plants (Table 1). So, these observations indicated that *Pseudomonas* culture filtrate can operate as a potential biocontrol agent for controlling root-knot nematodes. Similarly, the biocontrol efficiency of *Pseudomonas fluorescens* and *Pseudomonas protegens* Sneb 1997 is summed up in Table 1.

Chinheya et al. [41], under *in-vitro* conditions, tested 70 *Bacillus* isolates against *M. javanica* J2 on soybean. With serial dilutions, primary spore suspension was settled to 10^8^ per ml, and it was found that five isolates, BC 27, BC 29, BC 31, BC 56, and BC 64, caused mortality greater than 50%. From these five isolates, only three (BC 27, BC 29, and BC 31) from the rhizosphere of grass in goat pastures were chosen for second screening, as they caused greater larval mortality (80%) after 24 h. In a second in vitroscreening, it was found that BC 27 was remarkably better than BC 29 and BC 31, as it caused mortality of 100% after only 3 h, and BC 29 caused greater mortality than BC 31. After 24 h, both BC 27 and BC 29 were found to be more operative than BC 31 (Table 1). Under glasshouse experiments, in comparison to control, bacterial isolates BC 27 and BC 29 greatly reduced gall formation and the number of egg masses. The biocontrol potential of *Bacillus subtilis, Bacillus pumilis, Bacillus thuringiensis,* and *Bacillus altitudinis* is briefly summarized in Table 1.

*Bacillus* spp. have also been largely used for the effective management of plant-parasitic nematodes [46]. *Bacillus* spp. reduce the threat of chemical application by forming nematicidal metabolites [47,48]. *Bacillus subtilis*, a potential biocontrol agent, possesses spore-forming ability and several other characteristics that increases its chances of survival in the rhizosphere [49]. The genus *Pasteuria*, which includes the endospore-forming parasites, has also been found to decrease the populations of root-knot nematodes on various crops such as tomato, grapevines (*Vitis vinifera* L.), tobacco (*Nicotiana tabacum* L.) and peanut (*Arachis hypogaea* L.) [50]. It is a host-specific parasite of root-knot nematodes resistant to various nematicides and has a high level of virulence. Cetintas and Dickson [51] reported that in the presence of *Pasteuria penetrans*, the numbers of root galls by *Meloidogyne arenaria* race 1 were reduced on peanut. Similarly, Cho et al. [52] found that *Meloidogyne arenaria* was controlled by *Pasteuria penetrans* on tomato. Another bacterium, *Serratia plymuthica*, is found almost everywhere and can be used to control *M. incognita* (Table 1). It produces a large palette of antimicrobial products [35].

### 2.2. Fungi as Biocontrol Agents against Root-Knot Nematodes

Because fungi have a very high reproductive rate (both sexually and asexually), a short generation time, and are target-specific, the potential for the application of fungal biological control agents against plant pathogens has greatly expanded. Furthermore, in the absence of the host, they can survive in the environment by switching from parasitism to saprotrophism, allowing them to remain sustainable [53]. These fungi play an important role in controlling root-knot disease of plants caused by *Meloidogyne* spp. [54]. Different fungi have been found with nematophagous/nematicidal activities (Table 2). These fungi use different mechanisms to kill or control root-knot nematodes (Figure 2). More than 700 nematophagous fungi belonging to the phyla ascomycota, zygomycota, chytridiomycota, basidiomycota and oomycota have been described [55]. Nematophagous fungi were of four types: endoparasitic fungi, nematode-trapping fungi (predatory or scavengers), opportunistic ovicidal or parasites of eggs and females, and toxin-producing fungi [56,57].

#### 2.2.1. Entomopathogenic Fungi as Biocontrol Agents of Root-Knot Nematodes

Entomopathogenic fungi include those fungal species that parasitize other pathogens and kill them. *Beauveria bassiana* is an important entomopathogenic fungusthat is well-known to parasitize nematodes. Liu et al. [81] assessed the nematicidal activity of a culture filtrate of *B. bassiana* against *M. hapla* in *in-vitro* and greenhouse conditions on tomatoes. The percentage relative suppression rate of egg hatching ranged from 64.25 (50× dilution) to 99.89% (1× dilution), and mortality of J2 was directly proportional to the concentration of culture filtrate in vitro. In greenhouse conditions, soil drenching with 10×, 5×, 1× dilution strongly reduced the number of egg masses, root gall, and population density in tomato roots and soil. A 1× and 5× culture filtrate dilution resulted in 98.61% and 76.39% inhibition rate of nematodes, respectively. Youssef et al. [83] conducted *in-vitro* studies of the effect of culture filtrates at dilutions S, S/2, S/4 and conidial spore concentrations of 1 × 10^6^, 1 × 10^7^ and 1 × 10^8^ of the fungi *Beauveria bassiana*, *Metarhizium anisopliae* (Metchnikoff) Sorokin 1883 and *Purpureocillium lilacinus* on *M. incognita* egg hatching and juvenile mortality. *B. Bassiana and M. anisopliae* culture filtrate with dilution S resulted in maximum egg hatching inhibition by 90%, and *P. lilacinus* spore conc. 1 × 10^8^ resulted in the highest egg inhibition by 42.5%. The highest mortality of juveniles, of 100.00%, resulted from the culture filtrate of *M. anisopliae* at dilution S followed by 76% mortality by the fungus *P. lilacinus* at S dilution. In greenhouse conditions, *P. lilacinus* at S dilution resulted in the highest reduction in the number of egg masses (84.2%) and nematode populations in soil and root, by 86.4% and 82.9%, respectively. Additionally, the spore concentration (1 × 10^8^) achievedthe highest nematode reduction, by 85.3%. *B. bassiana* and *M. anisopliae* have been used as bioagents against root-knot nematodes, as they possess nematicidal activity. A secondary metabolite produced by *B. bassiana*, beauvericin, has toxic effects on *M. incognita* [84]. Ghayedi and Abdollahi [85] reported that the conidial spores of *M. anisopliae* attached to the cuticle of the nematode germinate, parasitize and produce infective hyphae inside the body of the nematode. Kershaw et al. [86] also reported the pathogenicity of the fungus due to the production of cyclopeptides and destruxins.

#### 2.2.2. Toxin-Producing Fungi as Biocontrol Agents against Root-Knot Nematodes

This group includes those fungal species that secrete some toxic compounds known as mycotoxins. A common example in this category is the oyster mushroom (*Pleurotus ostreatus*), which produces mycotoxin that causes paralysis of nematodes [87]. The toxin, trans-2-decenoic acid derived from linoleic acid, affects nematodes as well as insects and other fungi [88]. Five species of *Pleurotus,* including *P. ostreatus, P. sajor-caju* (Fr.) Fr. 1838, *P. cornucopiae* (Paulet) Rolland 1910, *P. florida* Cetto 1987 and *P.eryngii* (DC.) Quel. 1872, showed nematotoxic action against J2 of *M.javanica*. Toxin-producing processes or droplets were found on the hyphae. Contact of the nematode with toxin droplets resulted in immediate recoiling, and repeated contact resulted in inactivation. With increasing exposure (24–48 h), hyphae penetrated the nematode body, especially through the mouth, and within 2–3 days, their body contents were digested. *Pleurotus ostreatus* toxins were more effective, suddenly inactivating the nematodes after short exposure. Culture filtrates of *Pleurotus ostreatus* and *Pleurotus sajor-caju* showed similar toxicity [89]. Organic amendments with fresh mashed fruit residue of *Pleurotus ostreatus* at 15 g resulted in the reduction of 86.4% of nematodes and 92.4% of galls on cowpea and also increased plant growth and yield [77].

#### 2.2.3. Nematode-Trapping Fungi as Biocontrol Agents against Root-Knot Nematodes

A distinct and fascinating class of carnivorous microorganisms known as nematode-trapping fungi can capture and consume worms using specialized trapping structures. They are able to create a variety of trapping mechanisms, including constricting rings, non-constricting rings, adhesive knobs, adhesive networks, and adhesive hyphae [90]. *Arthrobotrys oligospora* Fresen. 1850 form different traps and adhesive devices for trapping and suppressing *M. incognita* second-stage juveniles (J2). The trapping organ consists of either a single ring or a fully developed 3-D network of hyphae. Mobile juveniles are trapped by the loop formed by hyphae, or hyphae directly adhere to immobilized juveniles without any developed loops or traps. Traps formed as a response against nematode presence and resulted in the attraction, adhesion, capturing, penetration and immobilization of juveniles and finally digestion and assimilation of the nematodes [67]. Studies revealed that trapping of juveniles by the fungus increases with time. The highest numbers of juveniles trapped were recorded after 72 h from nematode inoculation *in-vitro* [91]. Soliman et al. [66] carried out *in-vitro* and *in-vivo* experiments to study the effect of a culture filtrate of *A. oligospora* on *M. incognita*. The 50% and 100% concentrations of fungus culture filtrate resulted in 80.7% and 84.0% suppression of juveniles, respectively. In greenhouse experiments on tomato plants, soil applications of the fungus resulted in suppression of the numbers of females, galls and egg masses. In addition, treatment with the fungus prior to inoculation of nematodes showed a more significant effect in reducing the numbers of nematodes in soil and also resulted in increases in plant shoot weight, root weight and dry weight. Soil drenching with fungus alone resulted in an increase in plant growth, hence demonstrating the fungus to be a biocontrol against root-knot nematodes and an enhancer of plant vigor. Nourani et al. [92] studied the effect of two species of fungus, *A. oligospora* and *Arthrobotrys conoides* (Drechsler 1937), on egg hatching and mortality of *M. javanica* and *M. incognita*. Results showed that culture filtrates of *A. conoides* were more fatal to J2 and egg hatching as compared to *A. oligospora*. This was due to the production of the protein Ac1 by *A. conoides*, which could destroy a broad spectrum of substrate as compared to the serine protease produced by *A. oligospora* [93]. *M. incognita* juveniles were more sensitive to culture filtrate of both fungi as compared to *M. javanica.*

The nematode-trapping fungus *Aspergillus awamori* isolate BS05 parasitizes nematodes. The hyphae produced open constricting loops 18 h after incubation with nematodes, resulted in trapping by hyphal ring followed by penetration and digestion of the nematodes. In a pot experiment, treatment with fungal isolate resulted in reduction of root galling and eggs per root system and also improved tomato growth and yield [69]. *Dactylaria brochopaga*, a nematode-trapping fungus, formed traps for capturing larvae, paralyzed them, dissolvedthe outer cuticle, and digested the inner content, thus proving to be an effective antagonistic fungus for nematode management [94]. Noweer and Aboul-Eid [76] studied the effect of *D. brochopaga* for the management of *M. incognita* in *Cucumis sativus* L. cvs. Alfa. They observed that the fungus alone or in combination with yeast, molasses and vermiculite reduced nematode population density and the number of root galls and also significantly increased the weight of cucumber fruit per plant. *D. brochopaga* is also used a potent bioagent against *M. incognita* infesting superior grapevine [95]. Noweer et al. [75] suggested that combined application of *D. brochopaga* and *Verticillium chlamydosporium* (egg parasite of *Meloidogyne* spp.) with yeast molasses and vermiculate resulted in a 93.1% reduction of root galls per plant and also increased the weight of fruit per plant.

*Monacrosporium dermatum*, a nematode-trapping fungus, showed predatory efficacy against J2s of *M. incognita*. Trapping by the fungus increased with increases in the J2 population and incubation time; 64.50, 77.75, 91.50, 95.69, and 96.90% of juveniles were trapped in 100, 200, 300, 400 and 500 nematode population levels, respectively, at 96 h of exposure. Treatments of the fungus with organic manure on brinjal resulted in decrease in the nematode populations in root and soil and egg masses per plant, and also increased plant height and root length [96].

#### 2.2.4. Endoparasitic Fungi as Biocontrol Agents against Root-Knot Nematodes

A class of fungi known as endoparasitic fungi infects nematodes via their conidia. This class of fungi produces essentially no mycelium in soil and has no or a very limited saprophytic phase [97]. *Drechmeria coniospora* (Drechsler) W. Gams and H.B. Jansson, 1985is an endoparasitic nematophagous fungus used as a bioagent against nematodes. An extract from SDAY (10.0 g bacterial peptone, 10.0 g yeast extract, 40.0 g glucose, 18.0 g agar, 1 L water) medium offungal isolate of *D. coniospora* YMF1.01759 showed nematicidal activity against *M. incognita*, resulting in 99.1% mortality at 5 mg m/L at 72 h. EtOAc extract obtained from crude extract of SDAY attained 98.0% mortality of juveniles at 1 mg mL^−1^ at 72 h. From EtOAc extract, 13 metabolites were isolated and identified, and among them, 5-hydroxymethylfuran-2-carboxylic acid (1) at a concentration of 400 µg mL^−1^ exhibiteda toxic effect on *M. incognita*, resulting in 100% mortality of J2s after 72 h. The compound also inhibited egg hatching, as the number of hatched J2s from one egg mass was 11.17 at 200 µg mL^−1^ after 3 days [98].

#### 2.2.5. Ovicidal Fungi as Biocontrol Agents against Root-Knot Nematodes

The ovicidal group hunts and consumes eggs, cysts, and nematode females using traps and enzymes [99]. A good example in this category includes *Purpureocillium lilacinum* (Thom) Samson isolate HYBDPL-04, which has been reported to be effective against *M. incognita* among seven other isolates, NDPL-01, ANDPL-02, SHGPL-03, AHDPL-05, PTNPL-06, SNGPL-07 and VNSPL-08, resulting in inhibition of egg hatching by 90%, juvenile mortality of 80%, and 75% egg infection *in-vitro*. In a field experiment, application of HYBDPL-04 along with farmyard manure resulted in a reduction of galls by 60% and enhanced yield by 43% on tomato plants [100]. Field treatment with *P. lilacinus* 25%WP at 1.5 kg in 500 kg FYM first at transplantation and then after 30 days significantly reduced the nematode population by 24.50% in soil and induced a 60% yield enhancement [101]. Application of a *P. lilacinus* spore suspension (10 × 10^5^ conc.) to soil before transplantation of tomato plants reduced *Meloidogyne* spp. egg masses by 85% [59].

Bio-nematon, a biological nematicide of *P. lilacinus* formulation found to be very effective in reducing nematode populations of *M. incognita*, increases cucumber yield and is environmentally safe with no phytotoxic symptoms. Soil drenching first at the time of sowing and a second time 30–60 days after sowing with Bio-nematon *P. lilacinus* 1.15% WP bio-nematicide at 6.0 kg/ha was effective in managing *M. incognita* and increased plant yield [102]. Bio-nematon 1.15% at 69 g a.i/ha can also be effectively adopted for the management of root-knot nematodes (*M. incognita*) in tomato crops [103].

Bioformulations of the *P. lilacinus* (PI-181) strain in Attapulgite-based clay dust with peat colony count(CFU count 92.3%), demonstrated effective self-life(maximum spore viability of 94.9, 86.5, 81.2, 56.8 and 38.8% at 2, 4, 6 and 8 months, respectively), longevity and plant growth parameters and nematode reduction(76.72% egg infection) [104]. *P. lilacinus* 6029 filtrate obtained from Karanja cake-based broth was effective for the mortality of the J2 stage of *M. incognita*, as this cake medium elevated the production and nematicidal activity of the filtrate. Additionally, an increase in the incubation period of the culture filtrate of up to 15 days enhancedits nematicidal activity and resulted in 100% mortality of J2 within 12 h of exposure (*in-vitro* tests at the intervals of 3, 6, 12, and 24 h carried out in 24 well plates; each well contained 1 ml of culture filtrate and 0.5 ml suspension containing 150–200 juveniles) [105].

Bawa et al. [80] studied the integrated management of *M. incognita* in capsicum using *P. lilacinus* and organic amendments. They found that in a pot experiment, a bioformulations of *P. lilacinus* liquid 1.50% (1 × 10^9^ CFU/mL) at 6 L + FYM at 1.2 kg/500 kg soil was potent against the root-knot nematode and resulted in a 48.72% reduction in the average root gall index and 60.15% and 61.10% reduction in the average number of egg masses per root system and average soil nematode population, respectively. Hence, they carried out the experiment in a net house and in an open field with the same concentration (1.50%) of *P. lilacinus* along with organic amendments (neem cake and farmyard manure). It was observed that *P. lilacinus* liquid (1.50%) with enriched neem cake at 1 t/ha and FYM at 2.5 t/ha as a split application at the time of transplanting and 30 days after transplanting was very beneficial, as the treatment increased average shoot length (41.65 cm in net house and 35.13 cm in open field), shoot weight (57.86 g in net house and 43.02 g in open field), root length (13.17 cm in net house and 11.48 cm in open field) and root weight (13.47 g in net house and 10.35 g in open field). Application of *P. lilacinus* and neem also resulted in the reduction of the average root gall index by 44.52% and 50.60% and reduction of egg masses per root system by 56.92% and 61.08% in the net house and open field, respectively, compared to control.

Singh and Mathur [61] carried out an *in-vitro* study to assess the efficacy of the antagonistic fungi *Acremonium strictum*, *Aspergillus terreus*, *A. nidulans* G. Winter 1884, *A. niger*, *Cladosporium oxysporum* (Berk. and M.A. Curtis 1868), *Fusarium chlamydosporium* Link 1809, *F. dimarum*, *F. oxysporum* (Schlecht. Emend. Snyderand Hansen), *F. solani* (W.C. Snyderand andH.N. Hansen, 1941), *P. lilacinus*, *Pochonia chlamydosporia* (Goddard) Zare and W. Gams 2001, *Trichoderma viride* Pers 1794 and *T. harzianum* against *Meloidogyne incognita*. All fungi were tested for root-knot nematode egg parasitism, egg hatching reduction, and juvenile mortality. The highest percentage of egg parasitism was shown by *P. chlamydosporia* (87.2%), followed by *P. lilacinus* (82.3%), and *A. strictum* (53.3%) after 72 h of exposure of eggs to each of fungal culture, while no egg parasitism was shown by *A. terreus*, *A. nidulans*, *A. niger*, *F. dimarum,* and *F. solani*. Egg hatching was reduced by 11 to 14% by the culture filtrate of *A. strictum* (14.3%), *F. dimarum* (10.8%), and *F. oxysporium* (14.5%) after 10 days of exposure. The highest inactivation of juveniles was recorded by three fungi: *F. dimarum* (84.3%), followed by *A. terreus* (80.3%) and *A. strictum* (75.3%) after 72 h of exposure. The highest mortality of juveniles was caused by *A. strictum* (68.7%) and *A. terreus* (67.8%). *F. dimarum,* however, did not cause increased mortality of juveniles, and *F. solani* had no effect on nematode mobility or mortality of juveniles. The authors reported *A. strictum* and *A. terreus* as good bioagents against *M. incognita in-vitro*. *A. strictum* was found to be very effective in reducing egg hatching (14.3%), egg parasitism (53.3%) and inactivation (75.3%), and mortality (68.7%) of juveniles. It showed advantages over *P. chlamydosporium* and *P. lilacinus*, as it showed the dual activity of egg parasitism as well as toxin production. *In-vivo* applications of the fungi *Aspergillus terreus* and *Acremonium strictum* resulted in 76% and 73% reductions in eggs/egg mass and number of hatching eggs/egg mass by *A. terreus* and 71% and 68% by *A. strictum*, respectively. Combined application resulted in reductions in the egg hatching rate and number of eggs/egg mass by 77% and 81%, respectively.

#### 2.2.6. Secondary Metabolites and Nematicidal Compounds Obtained from Nematophagous Fungi as Biocontrol Agents against Root-Knot Nematodes

Various secondary metabolites from nematode-trapping fungi that exhibit nematicidal and nematode-attracting actions or influence the establishment of reproductive or trapping organs have been documented [106]. Some important metabolites with nematicidal activities include, Omphalotus olearius, linoleic acid, chaetoglobosin A, acetic acid, and oxalic acid [73]. *Aspergillus niger* strain F22 showed nematicidal activity in a dose-dependent manner against *M. incognita* and resulted in juvenile mortality and reduced egg hatching. A 5–20% culture filtrate resulted in complete suppression of egg hatching after 7 days of incubation and above 90% juvenile mortality with a 2.5–20% culture filtrate after 1 day of incubation. Analysis of culture filtrate revealed the presence of a major organic acid, oxalic acid, which showed nematicidal activity. A 2 mmol/L oxalic acid solution resulted in 100% mortality of *M. incognita* J2 after 1 day of treatment, and 10 and 50 mmol/L oxalic acid resulted in 100% egg hatching inhibition after 7 days of treatment. *M. hapla* was highly sensitive to oxalic acid in comparison to *M. incognita*. Oxalic acid resulted in the production of multiple vacuoles in the nematode body, which was straightened and stiffened, destroying all internal organs. The combined application of a 1:1 mixture of two wettable powder-type formulations of oxalic acid (90% purity) and *Aspergillus niger* F22 (spore count, 4.1 × 10^8^ CFU/g) at 1000- and 500-fold dilutions to a watermelon field reduced gall formation by 58.8% and 70.7%, respectively. Combined treatment showed greater inhibitory activity than a single treatment alone [107]. αβ-dehydrocurvularin (αβ-DC), a major nematicidal compound isolated from *Aspergillus welwitschiae* with a median lethal concentration (LC_50_) of 122.2 μg ml^−1^, showed nematicidal activity against *M. graminicola*. αβ-DC-treated rice root prevented attraction of nematodes, thus retarding invasion and infection. The volume of giant cells in treated roots was smaller than in the control; thus, the treatment had a negative impact on giant cell development, and this resulted in a decreased supply of nutrients to nematodes. Drenching with αβ-DC significantly reduced penetration by nematodes and thus reduced infection in rice root and suppressed the development of females. αβ-DC significantly reduced the root gall index under field conditions [108].

Volatile organic compounds(VOCs) obtained from various microorganisms have been used as bioagents for the control of nematodes [109]. Many VOCs produced by the fungus *Fusarium oxysporum*, including 2-methylbutyl acetate, 3-methylbutyl acetate, ethyl acetate, and 2-methylpropyl acetate, showed nematicidal activity against plant-parasitic nematodes [110]. VOCs obtained from the fungus *Daldinia* cf. *concentrica* (Bolton 1792) Cesati and de Notaris possess nematicidal activity against *M. javanica*, and 4-heptanone was its major active compound [111]. Mei et al. [112] studied the nematicidal potential of VOCs of the nematode-trapping fungus *Duddingtonia flagrans* (Dudd.) R.C. Cooke 1969. The fungal hyphae form a three-dimensional network trap for capturing and killing juveniles. Adhesive substances and mycelia also play a role in pathogenicity. GC-MS analysis of VOCs of the fungus revealed the presence of 52 metabolites, out of which cyclohexanamine, cyclohexanone, and cyclohexanol showed nematicidal activity. *M. incognita* mortality of 100% was achieved with 10 µL of cyclohexanamine and cyclohexanone at 12 h. Cyclohexanamine at 26.14 µM exhibited the highest inhibitory activity against egg hatching, resulting in 8.44 hatched juveniles per egg mass after 3 days.

Chitinases and proteases are essential extracellular enzymes involved in the degradation of chitinous eggshells [113]. Chitinase-producing fungi such as *Trichoderma harzianum, Lecanicillium lecanii* (R. Zare and W. Gams, 2001), and *Beauveria bassiana* have been used as biocontrol agents against plant-parasitic nematodes [114]. *P. lilacinus*, a nematophagous fungi that produces chitinase and protease, caused considerable damage to *M. javanica* egg shells [115]. CHI43, the first chitinase obtained from nematophagous fungi *Verticillium chlamydosporium* (Syn. *Pochonia chlamydosporia*) and *Verticillium suchlasporium* (Syn. *Pochonia rubescens* (Zare, W. Gams and Lopez-Llorca 2001), was effective against nematode eggshells [116].

Serine protease and PRA1 trypsin-like protease were isolated from *Trichoderma* species and showed nematicidal activity against juveniles of *M. javanica* [117]. Moreover, Aurovertins, pochonins and phomalactones obtained from *Pochonia chlamydosporia* showed egg parasitism and infected juveniles of nematode [118]. Grammicin, a metabolite isolated from *Xylaria grammica* (Mont. 1840) KCTC 13121BP showed nematicidal activity against *M. incognita,* egg hatching inhibition, and juvenile mortality at a 15.9 μg/mL concentration [119].

*Lecanicillium psalliotae* (Syn. *Verticillium psalliotae*) (Treschew 1941) Zare and W. Gams 2001,a filamentous nematophagous fungus used as a bioagent against root-knot nematodes, possesses a cuticle-degrading protease (Ver112) and chitinase (LPCHI1), which are effective against *M. incognita* eggs and thus influence its development [120,121]. Incubation of eggs of nematode with pure LPCHI1 and Ver112 resulted in the failure of 38.2% and 49.5% of the nematode eggs to hatch, respectively. The hatching rate was reduced by 56.55% using a combined treatment of both the chitinase and protease. Chitinase treatment resulted in deformed eggs, partially degraded eggshells and large vacuole formations in eggs [121]. *Lecanicillium antillanum* B-3 (Syn. *Verticillium antillanum* B-3) (R.F. Castaneda and G.R.W. Arnold) Zare and W. Gams 2001, a chitinolytic nematophagous fungus, showed parasitism on *Meloidogyne incognita* eggs. Treatment with B-3 isolate led to parasitization of 90% of the eggs by hyphae penetration, with growth inside the eggs on the third day. B-3 crude chitinase at a concentration of 14.6µg m/L resulted in damage to 78% of the eggs on day 4 of treatment [116]. *Lecanicillium muscarium*, a nematophagous fungus, parasitizes the eggs of *M. incognita* [121,122]. Hussain et al. [79] observed that the increased initial fungal conidia level decreased the nematode reproduction rate on tomato plants. Additionally, plant growth and yield increased with increasing fungus inoculum. A *Monacrosporium thaumasium* (Drechsler) de Hoog and Oorschot 1985 enzymatic pool containing protease proved effective in reducing the hatching rate of *M.javanica* eggs by 30% [123].

A fermentation filtrate of *Aspergillus japonicus* ZW1 showed nematicidal activity against *M. incognita.* Two-week fermentation filtrate(2-WF)showed strong nematicidal activity against the cumulative egg hatching rate as well as causing mortality of juveniles. With 50% 2-WF, the cumulative hatching rate of eggs was 6.4% after 15 days of incubation, and J2 mortality reached 100% after 6 h of incubation. Treatment with 2-WF resulted in wrinkles on the body surfaces of of juveniles, internal bubble formation and cytoplasmic vacuolization. In a greenhouse experiment, application of a 50% fermentation broth of 2-WF resulted in a 78.6–79.9% reduction in the number of root galls and a 69.4–72.0% reduction in the number of eggs per plant, compared to control. The nematicidal compound present in the fermentation filtrate was identified as 1,5-Dimethyl Citrate hydrochloride ester, which showed strong toxic activity against J2s. A 1.25 mg m/L concentration of this compound resulted in 91.7% mortality of J2s after 48 h of exposure. *A. japonicus* filtrate also showed no negative effect on seed germination of cucumber, tomatoes, wheat, rice, and cowpeas and thus proved to be a potent biocontrol agent [65].

*Aspergillus welwitschiae* AW2017 has ovicidal and larvicidal potential. A conidial suspension at a concentration of 5 × AW2017 resulted in 86.2% mortality of nematodes after 48 h of exposure, and egg hatching was reduced by 47.3% after 8 days of incubation. In addition, the conidial suspension of *A. welwitschiae* reduced the attractiveness of rice root to *M. graminicola* (Golden and Birchfield), resulting in a 64.9 ± 5.4% reduction in the number of penetrated juveniles and 39.6 ± 3.4% reduction the number of juveniles migrating to 3.0–4.5 cm. Drenching with the conidial suspension 5 × AW2017 resulted in reduction of galls and nematode penetration by 40.5% and 24.5%, respectively, and also delayed the development of nematodes in rice root [66].

*Chaetomium globosum*, a fungus living as saprophyte or as an endophyte, has been used as a bioagent against plant pathogenic microbes and nematodes [124]. Chaetoglobosin A (ChA) isa secondary metabolite of an isolate of *C. globosum* NK102 with anticancer activity and is an inhibitor of movement and mammalian cell proliferation. Natori [125] isolated it from its culture filtrate and also showed that it has nematicidal activity against *M. incognita*. *C. globosum* NK102 fungal colony, its culture filtrate and ChA all showed negative effects on juvenile mortality. *C. globosum* inhibited egg hatching but both filtrate and ChA showed no effect on eggs. The culture filtrate showed strong nematicidal activity toward juvenile mortality and penetration of J2s on cucumber root even at a very low 12.5% dilution. A 300 μg/mL concentration of ChA resulted in 99.8% mortality of J2s and also reduced penetration. Treatment of soil with 30 mg/kg of ChA significantly reduced the number of eggs per plant by 63% as compared to control [73]. In another isolate of *C. globosum* YSC5 culture broth, five secondary metabolites were isolated and showed adverse effects toward juvenile mortality. Among these metabolites, 200 μg/mL chaetoglobosin A(ChA), chaetoglobosin B(ChB) and flavipin resulted in 91.6, 83.8 and 87.4% mortality of J2s of *M. javanica*, respectively, after 72 h of exposure. ChA- and ChB-treated roots of tomato reduced nematode reproduction and also enhanced plant growth and yield. ChB at 200 μg/mL resulted in the reduction of galls by 61.5% and number of egg masses by 72.4%, and ChA caused 59.0% and 71.1% reductions, respectively, in the number of galls and egg masses [74].

### 2.3. Plant Extracts as Biocontrol Agents against Root-Knot Nematodes

The richest source of organic matter on Earth is found in plants, which are repositories of nature. These days, plants and their by-products are given more consideration as biocontrol agents against a variety of plant parasites, nematodes, fungi, and other pests [126]. According to current patterns in the application of botanical nematicides, a majority of the allelopathic substances demonstrate egg hatching inhibition, interrupt sexual selection, and decrease gut motility. Nematicidal plant extracts are mostly from the families Meliaceae, Fabaceae, Lamiaceae, Brassicaceae, Verbenance, Euphorbiaceae, etc. [127]. Various secondary metabolites released from plants such as alkaloids, flavonoids, glucosinolates, isothiocyanates, tannins, fatty acids, and sesquiterpenes show nematicidal potential against egg hatching, juvenile mortality, and penetration of nematodes [128,129]. Orisajo and Dongo [130] studied the nematicidal potential of various plant extracts of *Ocimum gratisimum* L., *Carica papaya* L., *Vernonia amygdalina* Delile, *Bixa orellana* L., and *Azadirachta indica* A. Jusson against the reproduction and pathogenicity of *Meloidogyne incognita*. Fabiyi [131] used plant materials of *Eucalyptus officinalis, Ocimum gratismum, Hyptis suaveolens* (L.) Kuntze and *Crotolaria juncea* L. as soil amendments to control *M. incognita* on okra. Cucurbitaceae cold peeling extracts (CCOPEs) obtained from the peels of *Cucumis melo* L. var. cantalupensis protected *Oryza sativa* L. and *Solanum lycopersicum* L. against root-knot nematodes by direct nematicidal effects and through resistance induced by the generation of reactive oxygen species (ROS) and ethylene accumulation and cell wall modification [132]. Arshad et al. [133] found that seed priming with botanical extracts, Neem (*Azadirachta indica*), Datura (*Datura stramonium* L.), Kortuma (*Citrullus colocynthis* (L.) Schrad) and *Moringa oleifera* Lam. leaf extracts showed significant results in suppressing nematodes. *In-vitro* neem leaf extract resulted in egg hatching reduction by 30% and juvenile mortality of 70%, and in pot trials, significantly reduced the numbers of galls, egg masses, and females and increased juvenile mortality. An aqueous extract of *Phyllanthus amarus* Schumach. and Thonn. (5000 ppm) resulted in 91% mortality of juveniles after 72 h of exposure and 86.5% reduction in egg hatching after 7 days of exposure [134]. Different plant parts, including by-products such as oil cakes, chopped leaves as soil or organic amendments, and plant extracts for soil drenching, seed dressing and priming have been assessed for their nematicidal properties: specifically, rhizome of *Curcuma longa* L., aerial parts of *Lantana camara* L., *Azadirachta indica*, *Datura*, and roots of *Fumaria parviflora* Lam., bark of *Terminalia nigrovenulosa* Pierre, 1886, and bulbs of *Allium sativum* L. Table 3 summarizes a few of the plant species, their secondary metabolites, and nematodes targeted.

## 3. Conclusions

Plant-parasitic nematodes, which cause huge economic losses globally, and sustainable and eco-friendly methods for their management are needed as harmful nematicides are being phased out from the agroecosystems. The necessity of such an alternate approach signifies the importance of biocontrol, which exploits the natural processes of various microorganisms, bacteria and fungi to control these parasitic nematodes. These biocontrol agents activate several processes that stimulate the self-regulation of the ecological community. They improve the nutrient content in soil, enhance its biological activities by augmenting it with beneficial microflora, promote plant growth, and exert lethal effects on root-knot nematodes through different modes including the production of metabolic compounds and other toxins that alter nematode egg hatching and kill nematode juveniles. However, efficient biocontrol requires detailed knowledge about the dynamics of pathogens and their natural enemies. In addition, the focus should be on combining more than one environmentally friendly discipline or approach for root-knot nematode management thatcan be effortlessly produced and easy to apply. Synergistic interaction, sustainability and other parameters as mentioned above must be improved for the ecofriendly management of nematodes.

## Figures and Tables

**Figure 1 plants-12-00451-f001:**
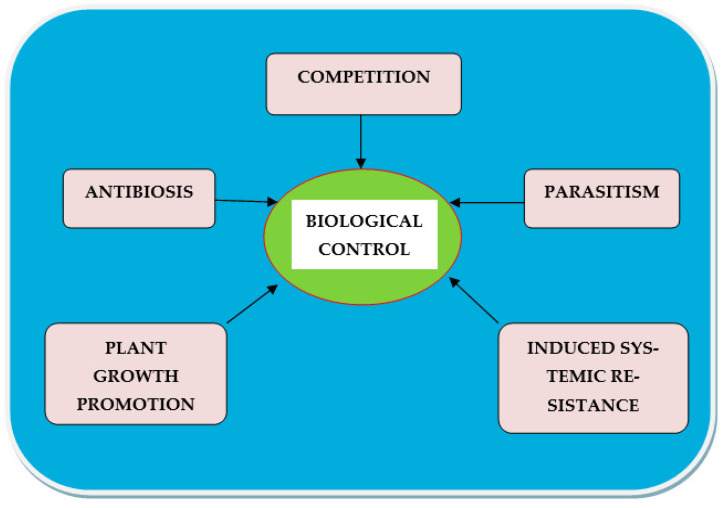
An overview of different mechanisms of biocontrol.

**Figure 2 plants-12-00451-f002:**
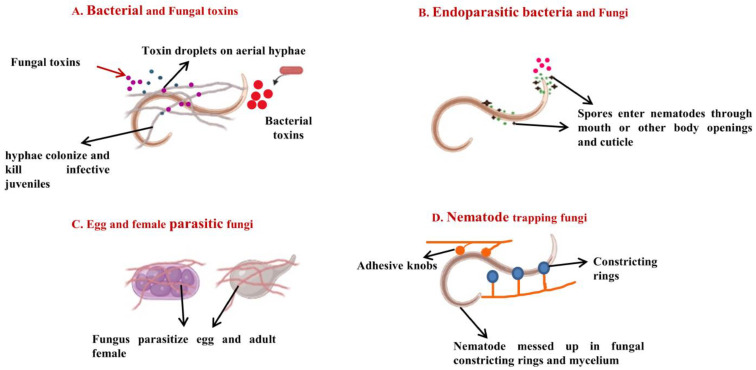
Mechanisms of bacterial and fungal biocontrol. (**A**). Bacterial and fungal toxins: Bacteria (e.g., *Bacillus thuringiensis*) and fungi (e.g., *Pleurotus ostreatus*) produce certain toxins that suppress plant-parasitic nematodes by preventing their hatching and may even cause death to juveniles [58]. (**B**). Endoparasitic bacteria and fungi: They produce motile spores that enter nematodes through the mouth or other body openings and cuticle and may be lethal to nematodes. (**C**). Egg- and female-parasitic fungi: Certain fungi such as *Pochonia chlamydosporia* parasitize the eggs and adult females of plant-parasitic nematodes [58]. They form branched mycelia around eggs and adult females and suppress plant-parasitic nematodes. (**D**). Nematode-trapping fungi: These fungi trap nematodes with the help of constricting rings or adhesive knobs and kill them, e.g., *Arthrobotrys dactyloides*.

**Table 1 plants-12-00451-t001:** Bacterial species as antagonists against root-knot nematodes.

Species Name	Concentration Used	Reduction in Diseases/Result	Crop	Nematode Managed	References
*Pseudomonas jessenii* Verhille, et al. 1999, and *Pseudomonas synxantha* (Ehrenberg 1840) Holland 1920	25%, 50%, 75%, 100%	All concentrations greater than 75% resulted in 100% mortality of J2.	Tomato (*Solanum lycopersicum* L.)	*Meloidogyne incognita*	[40]
*Bacillus* isolates (BC 27, BC 29, and BC 31)	10^8^ spores mL^−1^	BC 27 and BC 29 caused 100% mortality after 24 h. BC 31 was less effective compared to BC 27 and BC 29, as it caused only 84% mortality after 24 h.	Soybean (*Glycine max* (L.) Merr.)	*Meloidogyne javanica*	[41]
*Pasteuria penetrans* (ex Thorne 1940) Sayre and Starr 1986	50% spore suspension	Number of J2/100 cm^3^ was reduced to 9.2 in soil compared to 16.6 in control.	Babchi (*Psoralea corylifolia* L.)	*Meloidogyne incognita*	[42]
*Pseudomonas fluorescens* (pf1) (Flugge 1886) Migula, 1895	10^7^–10^9^ CFU/mL	69.8% reduction of *Meloidogyne incognita*	Cowpea (*Vigna unguiculata* (L.) Walp.)	*Meloidogyne incognita*	[43]
*Bacillus subtilis* (bs2) (Ehrenberg 1835) Cohn 1872	10^7^–10^9^ CFU/mL	82% reduction of total nematode population	Cowpea	*Meloidogyne incognita*	[43]
*Bacillus pumilis* (bp2) Meyer and Gottheli 1901	10^7^–10^9^ CFU/mL	81.8% reduction of nematode population	Cowpea	*Meloidogyne incognita*	[43]
*Bacillus thuringiensis* Berliner 1915	10^8^ CFU/mL/2	80.5% reduction of root-knot nematode	Tomato	*Meloidogyne incognita*	[44]
*Bacillus altitudinis* (AMCC1040) Shivaji et al. 2006	10^8^ CFU/mL	Numbers of J2s in roots and soil were reduced by 93.68% and 84.48%, respectively.	Ginger (*Zingiber officinale* Rosc.)	*Meloidogyne incognita*	[45]
*Pseudomonas protegens* Ramette et al. 2011	1 × 10^9^	Mortality rate of 87.76% was observed in J2 24 h after treatment (*in-vitro*), and Gall index was reduced to 30.67% compared to 49.33% in control, and biocontrol efficacy of 37.84 was observed	Tomato	*Meloidogyne incognita*	[35]
*Serratia plymuthica* (Lehmann and Neumann 1896) Breed et al. 1948	1 × 10^9^	Mortality rate of 92.67% was observed in J2 24 h after treatment (*in-vitro*), and Gall index lowered to 38.67% compared to 49.33% in control, and biocontrol efficacy of 21.62% was observed	Tomato	*M. incognita*	[35]

**Table 2 plants-12-00451-t002:** Fungal species for biocontrol of root-knot nematodes.

Species Name	Concentration Used	Reduction in Diseases/Result	Crop	Nematode Managed	References
*Purpureocillium lilacinus* Luangsa-ard, Houbraken, van Doom Hong Borman, Hywel-Jones, and Samson, 2011	Spore suspension, 10 × 10^5^ concentration	85% reduction of egg masses of *Meloidogyne* spp.	Tomato	*Meloidogyne* spp.	[59]
*Purpureocillium lilacinus*	1 × 10^9^ cfu/mL	76.24% reduction of root-knot diseases.	Tomato	*Meloidogyne incognita*	[60]
*Aspergillus terreus* Thom 1918, *Acremonium strictum* W. Gams 1971	2% (*w/w*) spore load, 2.3 × 10^6^ to 2.3 × 10^8^	76% and 73% reduction in eggs/egg mass and the number of hatching egg/egg mass by *A. Terrus,* and 71% and 68% by *A. Strictum,* respectively.	Tomato	*Meloidogyne incognita*	[61]
*Acremonium implicatum* (J.C. Gilman and E.V. Abbott) W. Gams, 1975	Spore suspension, 1 × 10^6^ CFU/mL	Reduction in galls with 40.6 galls/treated plant as compared with 121.6 on control plant.	Tomato	*Meloidogyne incognita*	[62]
*Acremonium implicatum*	1 × 10^6^ CFU/mL conidial suspension	Reduction of 60% root galls	Tomato	*Meloidogyne incognita*	[63]
*Acremonium strictum, Aspergillus niger* van Tieghem 1867, *Purpureocillium lilacinus* and *Trichoderma harzianum* Rifai 1969	Talc-based formulation with spore load 2 × 10^8^ CFU	Combined effect of *Trichoderma harzianum* and *Acremonium strictum* resulted in greater reduction of disease and high yield.	Tomato	*Meloidogyne incognita*	[64]
*Arthrobotrys dactyloides* Drechsler 1937	2 g (10,000 spore concentration) + 1 g yeast + 3 g vermiculite + 1 mL molasses	Reduction of 94.1% of root galls per plant	Snap bean (*Phaseolus vulgaris* L.)	*Meloidogyne incognita*	[65]
*Arthrobotrys oligospora* Fresen 1850	Spore suspension with 10^5^ conidia/mL	Reduced number of galls, females and nematodes and enhanced plant growth.	Tomato	*Meloidogyne incognita*	[66]
*Arthrobotrys oligospora*	Fungal suspension at 10, 30 and 50 mL/plant (1 × 10^4^ spore/mL)	50 mL/plant resulted in reduction of females, eggs/egg-mass and no. of J2 in soil.	Tomato	*Meloidogyne incognita*	[67]
*Arthrobotrys oligospora*	10^6^ spores/mL	Application of fungus with salicylic acid reduced root galls and nematode population and increased plant growth.	Tomato	*Meloidogyne javanica*	[68]
*Aspergillus awamori* Nakaz	10^8^ CFU/mL	Resulted in 44.9% reduction of nematode infection.	Tomato	*M. incognita*	[69]
*Aspergillus japonicus ZW1* Saito 1906	20% fermentation broth	Resulted in 51.8 and 47.3% reduction of eggs and galls, respectively.	Tomato	*M. incognita*	[70]
*Aspergillus welwitschiae* AW2017 (Bres.) Henn.	2 × 10^8^ conidia/mL (5× AW2017)	Reduction by 40.5% and 24.5% of root galls and juveniles, respectively.	Rice (*Oryza sativa* L.)	*M. graminicola*	[71]
*Fusarium* and *Trichoderma* isolates	5 × 10^6^ conidial suspension per pot	29–42% of root galling was reduced by application of conidia of rhizosphere *Fusarium* isolates and 38% reduction of root galls by treatment with *Trichoderma*.	Rice	*M. graminicola*	[72]
*Chaetomium globosum* Kunze 1817	30 mg ChA/kg soil (Chaetoglobosin A-ChA)	Resulted in reduction of 63% of eggs per plant	Cucumber (*Cucumis sativus* L.)	*M. incognita*	[73]
*Chaetomium globosum YSC5*	200 μg/Ml of chaetoglobosin B and chaetoglobosin A	59.0–61.5% reduction in number of galls and 71.1–72.4% reduction in number of egg masses	Tomato	*M. javanica*	[74]
*Dactylaria brochopaga* Drechsler 1937 & *Verticilium chlamydosporium* Goddard 1913	2 g (*Dactylaria* + *Verticilium chlamydosporium*) + 3 g (vermiculite) + 1 mL (molasses) + 1 g yeast	93.1% reduction of root galls per plant	Eggplant (*Solanum melongena* L.)	*M. incognita*	[75]
*Dactylaria brochopaga*	2 g (fungus) + 1 g (yeast) + 1 mL (molasses) + 3 gm (vermiculite)	Resulted in 94.1% mean reduction in the number of root galls	Cucumber	*M. incognita*	[76]
*Pleurotus ostreatus* (Jacq.) P. Kumm. 1871	5, 10, 15 g fresh mashed mushroom	15 g mushroom residue resulted in an 86.4% reduction of nematode reproduction and gall reduction by 92.4%.	cowpea	*M. incognita*	[77]
*Gliocladium* spp.	10^4^ mL^−1^, 10^5^ mL^−1^, 10^−6^ mL^−1^ conidia suspension	10^6^ mL^−1^ conidia suspension significantly decreased intensity of damage by 33%.	Tomato	*Meloidogyne* spp.	[78]
*Lecanicillium muscarium* R. Zare and W. Gams 2001	10^3^, 10^4^, 10^5^ and 10^6^ conidia levels with different inoculum densities of *M. incognita* (500, 1000, 1500, 2000)	Higher density 1 × 10^6^ decreased nematode population, and plant growth parameters improved with increasing fungus inoculum.	Tomato	*M. incognita*	[79]
*Purpureocillium lilacinus*	*P. lilacinus* WP 1.15% (1 × 10^8^ CFU/g), *P. lilacinus* liquid 1.50% (1 × 109 CFU/mL) and *P. lilacinus* AS 1.0% (2 × 10^6^ CFU/g)	*P. lilacinus* liquid 1.50% resulted in 48.72% reduction in average root gall index; average number of egg masses per root system and average soil nematode population reduced by 60.15% and 61.10%, respectively.	Capsicum (*Capsicum annuum* L.)	*M. incognita*	[80]
*Beauveria bassiana* (Bals. Criv.) Vuill. 1912	1×, 5×, 10×, 20×, 50× dilution of culture filtrate	Resulted in 98.61% and 76.39% rates of inhibition of nematodes at 1× and 5× solutions	Tomato	*M. hapla*	[81]
*Arthrobotrys dactyloides* Drechsler 1937	4 × 10^6^ CFU/kg of soil	Resulted in reduction of 37.9–81.8% of juveniles and 44.5–51.3% of egg masses	Tomato	*M. incognita*	[82]

**Table 3 plants-12-00451-t003:** Plant extracts for biocontrol of root-knot nematodes.

Plant Species	Family	Plant Parts Used	Active Compounds	Nematodes Targeted	References
*Azadirachta indica*	Meliaceae	Leaf, Seed, Fruit, Root, Bark	Azadirachtin	*Meloidogyne incognita, M. javanica*	[135,136]
*Lantana camara*	Verbenaceae	Aerial part	Lantanilic acid, camaric acid and oleanolic acid	*M. incognita*	[137]
*Fumaria parviflora*	Papaveraceae	Root	Nonacosane-10-ol and 23a-homostigmast-5-en-3β-ol	*M. incognita*	[138]
*Tageteserecta* L.	Asteraceae	Leaves	Alpha-terthienyl	*M. incognita, M. javanica*	[139,140,141]
*Moringa oleifera*	Moringaceae	Leaves	Flavonoids, glycosides, saponin	*M. incognita*	[142]
*Terminalia nigrovenulosa*	Combretaceae	Bark	3,4-dihydroxybenzoic acid (3,4-DHBA)	*M. incognita*	[143]
*Datura* spp.	Solanaceae	Leaves, inflorescence, roots	Atropine, scopolamine, hyoscyamine	*Meloidogyne* spp.	[144]
*Juglans regia* L.	Juglandaceae	Leaves, husk	Beta 1, 4 naphthoquinones	*M. hispanica, M. luci*	[145,146]
*Waltheria indica* L.	Malvaceae	Roots	5-methoxywaltherione A, waltherione A and waltherione C	*M. incognita, M. hapla, M. arenaria*	[147]
*Hedysarum coronarium* L.	Fabaceae	Leaves, flower	Saponins, flavonoids and tannins	*M. incognita*	[148]
*Allium sativum* L.	Alliaceae	Bulb, leaves	Organosulfur compound, Allicin	*Meloidogyne* spp.	[149,150]
*Cymbopogon martini* (Roxb.) Watsand *C. flexuosus* (Nees ex Steud) W. Watson	Poaceae	Leaves	Eugenol and citral	*M. incognita*	[151]
*Brassica* spp.	Brassicaceae	Shoot, roots, seed	Isothiocynates	*Meloidogyne* spp.	[152]
*Pistacia lentiscus* L.	Anacardiaceae	Leaves	Quercetin, quinic and gallic acid	*M. javanica*	[153]

## Data Availability

Not applicable.

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
