# Peer review of "Microbes vs. Nematodes: Insights into Biocontrol through Antagonistic Organisms to Control Root-Knot Nematodes"

_plants, 2023, doi:10.3390/plants12030451_

Round 1

Reviewer 1 Report

Greetings and respect from the writer and editor of the magazine.

In the introduction, a complete explanation of the process is provided.

Main article question:

Microbes vs Nematodes: Insights into Biocontrol through Antagonistic Organisms to Control

The article correctly describes the topic and has the necessary innovation.

Due to EU regulations on the use of chemical nematicides, it is now very important to explore environmentally friendly control strategies for nematode management.

Figure 2 needs a clearer explanation.

The research language needs to be corrected grammatically.

Finally, with some changes, the article is accepted.

Author Response

Thanks for the valuable comments. Changes have been made to the manuscript as suggested.

Figure 2 has been revised with more clear explanations.

Research language has been modified throughout the manuscript.

Reviewer 2 Report

Please carefully edit this manuscript and resubmit. I have found corrections on almost every line and stopped my review due to numerous errors in style.

This paper needs to be better organized and the tables and figures need to be better explained.

Author Response

Thank you very much for providing all the scientific modifications in the manuscript. All the comments have been addressed and response is attached as word file. 

Round 2

Reviewer 2 Report

Page 1, Line 35: Root-knot nematodes do not transmit viral diseases nor do they acquire viruses when they feed on virus infected plants. Instead, they do interact with viral diseases in several ways: 

From Weischer, Bernhard,    Nematode-virus interactions. Pp. 217-231, in M. Wajid Khan ed. Nematode Interactions. Springer: Suffolk.

“In Solanum khasium the root-knot index was higher on plants inoculated with tobacco mosaic tobamovirus than on healthy plants (Ismail et al., 1979). Inhibitory effects on root-knot nematode M. javanica were observed in zucchini (Cucurbita pepo) infected with watermelon mosaic potyvirus. Virus infection retarded the establishment of these nematodes in the roots as compared with healthy plants (Huang and Chu, 1984). All these effects, favourable or detrimental, occurred or were more pronounced when nematode inoculation was preceded for two to three weeks by virus infection. This points to biochemical processes in plant tissues as a decisive mechanism involved in these interactions. This is underlined by results obtained by Alam et al. (1990). They observed antagonistic effects between tomato mosaic tobamovirus and Mincognita. When virus infection preceded nematode inoculations nematodes were suppressed, and when nematodes were the first agent the virus was inhibited. The changes caused by one pathogen were detrimental to the other. The influence of the host on this type of interaction was demonstrated by Moura and Powell (1977). In two out of three tomato varieties the egg production of Mincognita was significantly increased by the presence of tobacco mosaic tobamovirus but in the third it was not. The concomitant presence of nematode and virus in most cases results in a synergistic effect that aggravates the plant damage considerably. Little is known about the physiological or biochemical basis of enhancing or inhibiting nematode development by a virus infection of the host. It has been suggested that changes in protein metabolism caused by nematodes may interact with corresponding changes caused by the virus, resulting in beneficial effects in one case and detrimental effects in another (Weischer, 1975). This view was recently supported by ShowIer et al. (1990). They showed that the changes in free amino acid levels in sugarcane caused by sugarcane mosaic virus were responsible for population changes in various nematodes on sugarcane.”

Page 3, Figure 1: reverse the arrows to point toward “Biological Control” 

Author Response

We are thankful to the reviewers for their deep and thorough review. We have revised the manuscript in light of their useful suggestions and comments. We hope the revisions have improved the paper to a level to their satisfaction. All the changes have been marked in red.

Page 1, Line 35: Root-knot nematodes do not transmit viral diseases nor do they acquire viruses when they feed on virus infected plants. Instead, they do interact with viral diseases in several ways: 

From Weischer, Bernhard,    Nematode-virus interactions. Pp. 217-231, in M. Wajid Khan ed. Nematode Interactions. Springer: Suffolk.

Response: thank you for the valuable suggestion. The changes are incorporated in the revised manuscript as suggested.

Page 3, Figure 1: reverse the arrows to point toward “Biological Control” 

Response: corrected in the revised manuscript

The manuscript has been briefly revised and the suggested changes has been made and highlighted.

Thank you so much for your time and concentration. We hope these improvements made to the manuscript and the above explanations are adequate and clear and hope this manuscript will be more acceptable for publication.